# Learning to perceive objects by prediction

**Tushar Arora**
IRCN, The University of Tokyo
arora.tushar1208@gmail.com

**Li Erran Li**
Alexa AI, Amazon
erranli@gmail.com

**Ming Bo Cai**
IRCN, The University of Tokyo
mingbo.cai@ircn.jp

## Abstract

The representation of objects is the building block of higher-level concepts. Infants develop the notion of objects without supervision. The prediction error of future sensory input is likely the major teaching signal for infants. Inspired by this, we propose a new framework to extract object-centric representation from single 2D images by learning to predict future scenes in the presence of moving objects. We treat objects as latent causes of which the function for an agent is to facilitate efficient prediction of the coherent motion of their parts in visual input. Distinct from previous object-centric models, our model learns to explicitly infer objects' locations in a 3D environment in addition to segmenting objects. Further, the network learns a latent code space where objects with the same geometric shape and texture/color frequently group together. The model requires no supervision or pre-training of any part of the network. We created a new synthetic dataset with more complex textures on objects and background and found several previous models not based on predictive learning overly rely on clustering colors and lose specificity in object segmentation. Our work demonstrates a new approach for learning symbolic representation grounded in sensation and action.

## 1 Introduction

Visual scenes are composed of various objects in front of backgrounds. Discovering objects from 2D images and inferring their 3D locations is crucial for planning actions in robotics (Devin et al., 2018; Wang et al., 2019) and this can potentially provide better abstraction of the environment for reinforcement learning (RL), e.g. Veerapaneni et al. (2020). The appearance and spatial arrangement of objects, together with the lighting and the viewing angle, determine the 2D images formed on the retina or a camera. Therefore, objects are latent causes of 2D images, and discovering object is a process of inferring latent causes (Kersten et al., 2004). The predominant approach in computer vision for identifying and localizing objects rely on supervised learning to infer bounding boxes (Ren et al., 2015; Redmon et al., 2016) or pixel-level segmentation of objects (Chen et al., 2017). However, the supervised approach requires expensive human labeling. It is also difficult to label every possible category of objects. Therefore, an increasing interest has developed recently to build unsupervised or self-supervised models to infer objects from images, such as MONet (Burgess et al., 2019), IODINE (Greff et al., 2019) slot-attention (Locatello et al., 2020), GENESIS (Engelcke et al., 2019, 2021), C-SWM (Kipf et al., 2019) and mulMON (Nanbo et al., 2020). Our work also focuses on the same unsupervised *object-centric representation learning* (OCRL) problem, but offers a new learning objective and architecture to overcome the limitation of existing works in segmenting more complex scenes and explicitly represents objects' 3D locations.

3rd Workshop on Shared Visual Representations in Human and Machine Intelligence (SVRHM 2021) of the Neural Information Processing Systems (NeurIPS) conference, Virtual.

The majority of the existing OCRL works are demonstrated on relatively simple scenes with objects of pure colors and background lacking complex textures. As recently pointed out, the success of several recent models based on a variational auto-encoder (VAE) architecture (Kingma & Welling, 2013; Rezende et al., 2014) depends on a reconstruction bottleneck that needs to be intricately balanced (Engelcke et al., 2020). To evaluate how such models perform on scenes with more complex surface textures, we created a new dataset of indoor scenes with diverse texture patterns on the objects and background. We found that several existing unsupervised OCRL models overly rely on clustering pixels based on their colors. A challenge in our dataset and in real-world perception is that sharp boundaries between different colors exist both at contours of objects and within the surface of the same object. A model essentially has to implicitly learn a prior knowledge of which types of boundaries are more likely to be real object contours. The reconstruction loss in existing works appears to be insufficient for learning this prior.

To tackle this challenge, we draw our inspiration from development psychology and neuroscience. Infants understand the concept of object as early as 8 months, before they can associate objects with names (Piaget & Cook, 1952; Flavell, 1963). The fact that infants are surprised when object are hidden indicates that they have already learned to segment discrete objects from the scene and that their brains constantly make prediction and check for deviation from expected outcome. As the brain lacks direct external supervision for object segmentation, the most likely learning signal is from the error of this prediction. In the brain, a copy of the motor command (efference copy) is sent from the motor cortex simultaneously to the sensory cortex, which is hypothesized to facilitate prediction of changes in sensory input due to self-generated motion (Feinberg, 1978). What remains to be predicted are changes in visual input due to the motion of external objects. Therefore, we hypothesize that one functional purpose of grouping pixels into object is to allow the prediction of the motion of the constituting pixels in a coherent way by tracking very few parameters (e.g., the location, pose, and speed of an object). Driven by this hypothesis, our contribution in this paper is: (1) we combine predictive learning and explicitly 3D motion prediction to learn 3D aware object-centric representation; (2) we found several previous models overly rely on clustering colors to segment objects; (3) although our model leverages image prediction as learning objective, the architecture generalize the ability of object segmentation and spatial localization to single-frame images.

## 2 Method

### 2.1 Problem formulation and Network architecture

We consider each scene to have $k$ distinct objects and a background. At any moment $t$, based on an object's 3D location and pose relative to the observer (camera), a 2D image is rendered on the camera. Our goal is to train a neural network $f_{\text{obj}}$ that infers properties of objects given only a single image $\boldsymbol{I}^{(t)}$ as the sole input without external supervision. The network would sequentially output a set of view-invariant vectors representing the identity of each object and a vector for each pixel denoting its probability of belonging to any of the K objects or the background which acts as our segmentation mask. Alongside, the network estimates the location and a vector describing the probabilistic distribution of each object's pose (yaw) relative to the observer.

Our hypothesis is that the notion of object emerges to meet the need of efficiently predicting the future fates of all parts of an object. With the (to be learned) ability to infer an object's pose and location from each frame, the object's speed of translation and rotation can be estimated from consecutive frames. We can further predict optical flow of each pixel based on object's speed, inferred depth, and pixel position relative to the object's center. The pixel-segmentation of an object essentially prescribes which pixels should move together with the object. With the predicted optical flow, one can further predict part of the next image by warping the current image. The parts of the next image unpredictable by warping include surfaces of objects or the background that are currently occluded but will become visible, and the region of a scene newly entering the view due to self- or object-motion. These portions can only be predicted based on the learned statistics of the appearance of objects and background, which we call "imagination". We show that with the information of self-motion, knowledge of geometry (rule of rigid-body motion) and the assumption of smooth object movement (speed and rotation does not change in the next frame), the object representations captured by function $f_{\text{obj}}$ and depth perception can be learned without supervision.

We name our model Object Perception by Predictive LEarning (OPPLE) network. Our model has 3 networks as displayed in Figure 1. **Depth Perception Network** is a standard U-Net for predicting the depth for each input image. **Object Extraction Network** is a modified U-Net, in which the output of the encoder is repeatedly read by an LSTM to predict object location and pose distribution as well as a view-invariant object representation code. This view invariant representation is then decoded to generate a probabilistic mask over each image. **Object-based Imagination network** is also a modified U-Net which takes an image and its estimated depth weighted by the mask output of the Object Extraction Network as input along with the observer's and object's motions to predict the RGB values and depth for the regions that were occluded by an object before but should be visible to the observer after the observer and objects' motion from time $t$ to $t + 1$. Please note that imagination here does not indicate creating appearance entirely from scratch but predicting occluded parts based on visible parts. More detailed formulation and the pseudocode of the algorithm is provided in the appendix.

To train these three networks jointly, we optimize the loss function between the RGB values of the *predicted* and *actual* image for time point $t + 1$, with the predicted image produced by warping the image at time $t$ based on the predicted optical flow and the imagined image in the occluded regions. We also calculate loss between the inferred depth by the depth prediction network and the predicted depth by the same warping approach for the image in log scale. Additional spatial regularization includes the distance between the predicted and inferred object location and the KL divergence between the predicted and inferred pose for objects. More details for the loss function and network architecture are provided in the appendix.

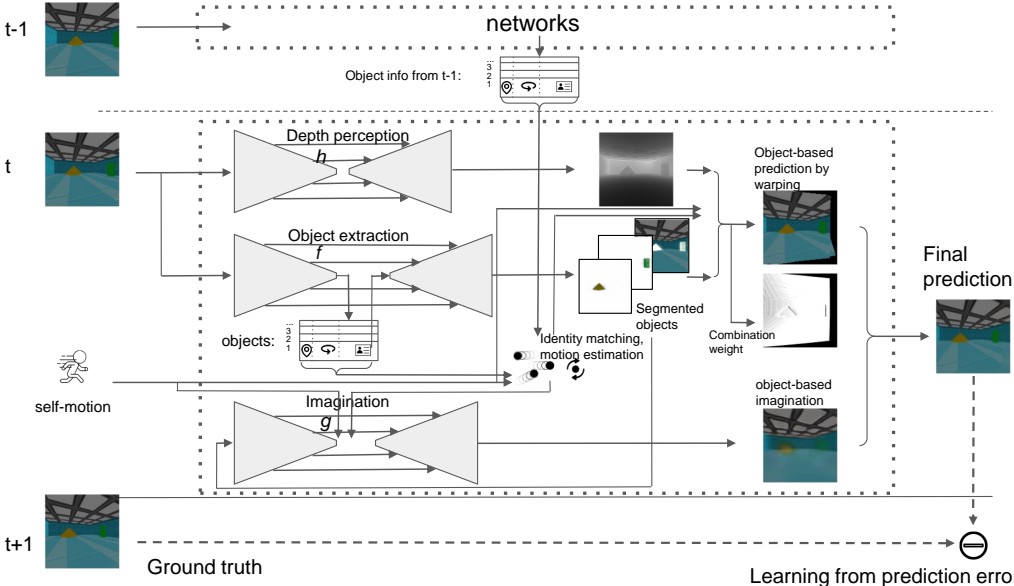

Figure 1: **Architecture for the Object Perception by Predictive LEarning (OPPLE) network** Images at each time point are processed by the Object Extraction and Depth Perception Networks independently. All networks use U-Net structure. The dotted boxes indicate the entire OPPLE network acting at each time step (details omitted for $t - 1$). Motion information of each object is estimated from the spatial information extracted for each object between $t - 1$ and $t$. Objects between frames are soft-matched by a score depending on the distance between their latent codes. Self and object motion information are used together with object segmentation and depth map to predict the next image by warping the current image. The segmented object images and depth, together with their motion information and the observer's motion, are used by the imagination network to imagine the next scene and fill the gap not predictable by warping. The error between the final combined prediction and the ground truth of the next image provides teaching signals for all three networks.

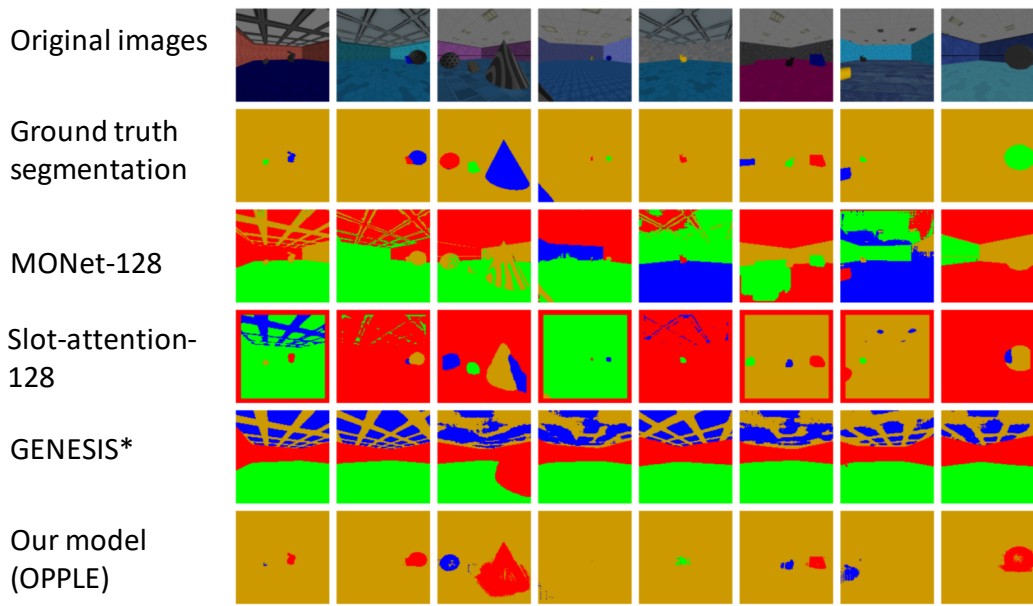

Figure 2: Example of object segmentation by different models

# 3 Results

After training the networks, we evaluate them on 4000 test images from the same distribution. We compare different models mainly on their segmentation performance. Additionally, we demonstrate the ability unique to our model: inferring locations of objects in 3D space and the depth of the scene.

## 3.1 Object segmentation comparison

Following prior works (Greff et al., 2019; Engelcke et al., 2019, 2021), we evaluated segmentation with the Adjusted Rand Index of foreground objects (ARI-fg). In addition, for each image, we matched ground-true objects and background with each of the segmented class by ranking their Intersection over Union (IoU) and quantified the average IoU over all foreground objects.

Our model outperforms most compared models in ARI-fg and is only second to a bigger slot-attention network in IoU. It is shown qualitatively in Figure 2 and quantitatively in Table 3.1, MONet and GENESIS appear to heavily rely on color to group pixels into the same masks. Slot-attention performs well in segmenting objects but still treats patterns of ceiling as additional objects potentially due to bias on clustering colors. Even though some of these models almost fully designate pixels of an object to a mask, the masks lack specificity in by including pixels with similar colors from other objects or background. Patterns on the backgrounds are often treated as objects as well. The reason of such drawbacks awaits further investigation but we postulate

| Model | ARI-fg | IoU |
|---|---|---|
| MONet | 0.31 | 0.08 |
| MONet-128 | 0.33 | 0.22 |
| MONet-128-bigger | 0.33 | 0.15 |
| slot-attention | 0.41 | 0.31 |
| slot-attention-128 | 0.39 | **0.54** |
| GENESIS | 0.17 | 0.03 |
| our model (OPPLE) | **0.46** | 0.35 |

Table 1: Performance of different models on object segmentation.

there may be fundamental limitation in the approach that learns purely from static discrete images. Patches in the background with coherent color offer room to compress information similarly as objects with coherent colors do, and their shapes re-occur across images just as other objects. This may incentivize the networks to treat them as objects. Our model is able to learn object-specific masks because these masks are used to predict optical flow specific to each object. A wrong segmentation would generate large prediction error even if the motion of an object is estimated correctly. Such prediction error forces the masks to be concentrated on object surface. They emerge first at object boundaries where the prediction error is the largest and gradually grow inwards during training.

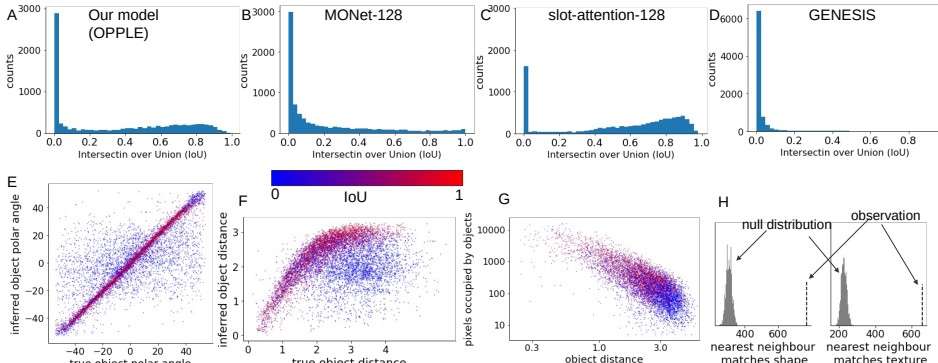

Figure 3: **A-D**: distribution of IoU. All models have IoU < 0.01 for about 1/4 of objects. Only OPPLE shows a bi-modal distribution while other models' IoU are more skewed towards 0. **E-F**: object localization accuracy of OPPLE for object's polar angle and distance relative to the camera. Each dot is a valid object with color representing its segmentation IoU. Angle estimation is highly accurate for well segmented objects (red dots). Distance is under-estimated for farther objects. **G**: objects with failed segmentation (blue dots) are mostly far away and occupying few pixels. **H** The numbers of objects sharing the same shape or texture with their nearest neighbour objects in latent space are significantly above chance.

Figure 3A-D further compares the distribution of IoU across models. Most models have IoU < 0.01 for about 1/4 of objects. Only OPPLE and slot attention show bi-modal distributions while other models' IoU are more skewed towards 0. Figure 3G plots each object's distance and size on the picture with colors corresponding to their IoUs in our model. Objects with poor segmentation (blue dots) are mostly far away from the camera and occupy few pixels. This is reasonable because motion of farther objects causes less shift on the images and thus provide weaker teaching signal for the network to attribute their pixels as distinct from the background.

## 3.2    Object localization

One quantity inferred by the Object Extraction Network is each object's location relative to the camera, represented as the bearing angle and distance in polar coordinate relative to the camera. Figure 3E-F plot the true and inferred angles and distance, color coded by objects' IoUs. For objects well segmented (red dots), their angles are estimated with high accuracy (concentrated on the diagonal in E). Distance estimation is negatively biased for farther objects, potentially because the regularization term on the distance between the predicted and inferred object location at frame $t + 1$ favors shorter distance when estimation is noisy. Poorly localized objects are mainly those with poor segmentation (blue dots). Note that the ability to explicitly infer object's location is not available in other models compared.

## 3.3    Meaningful latent code

Because a subset of the latent code (10 dimensions) was used to calculate object matching scores between frames in order to soft-match objects, this should force the object embedding $z$ to be similar for the same objects. We explored the geometry of the latent code by examining whether the nearest neighbours of each of the object in the test data with IoU > 0.5 are more likely to have the same property as themselves. 772 out of 3244 objects' nearest neighbour had the same shape (out of 11 shapes) and 660 objects' nearest neighbour had the same color or texture (out of 15). These numbers are 28 to 29 times the standard deviation away from the means of the distribution expected if the nearest neighbour were random (Figure 3H). This suggests the latent code reflects meaningful features of objects. However, texture and shape are not the only factors determining latent code, as we found the variance of code of all objects with the same shape and texture to still be big.

## 3.4    Performance on depth perception

We demonstrate a few example images and the inferred depth. Our network can capture the global 3D structure of the scene, although details on object surfaces are still missing. Because background

occurs in every training sample, the network appears to bias the depth estimation on objects towards the depth of the walls behind, as is also shown in the scatter plot.

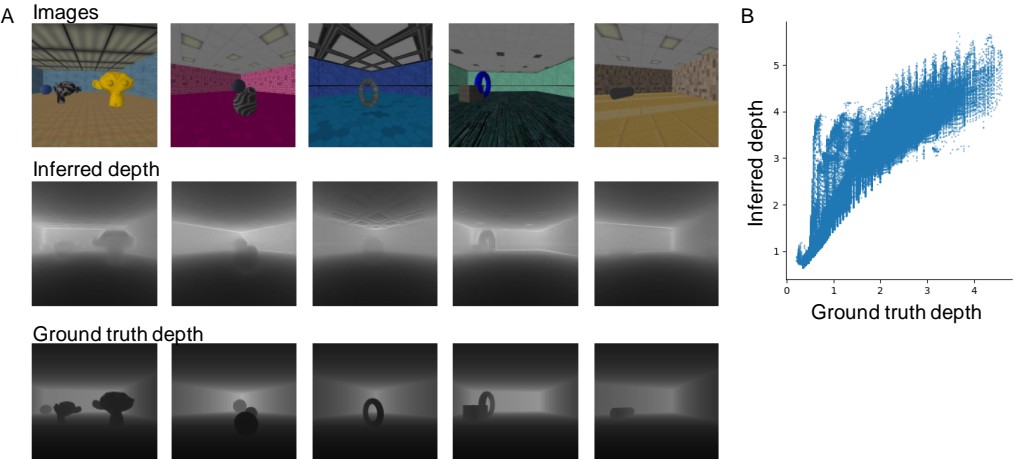

Figure 4: **A** Comparison between ground truth depth and inferred depth. **B** Shows the variation of Inferred depth with respect to the ground truth depth.

## 4 Discussion

We provide a new approach to learn object-centric representation that includes explicit spatial localization of objects, object segmentation from image, automatic matching the same objects across scenes based on a learned latent code and depth perception as a by-product. All of the information extracted by our networks are learned without supervision and no pre-training on other tasks is involved. The only additional information required is that of observer's self-motion, which is available both in the brain as efference copy and accessible in vehicles (although likely inaccurate). Our work is inspired by the seminal theoretical work of (O'Reilly et al., 2021), while we move beyond and designed the architecture to allow learning properties of objects more explicitly (segmentation and localization). The work demonstrates that the notion of object can emerge as a necessary common latent cause of the pixels belonging to the object for the purpose of efficiently explaining away the pixels' coherent movement across frames. This illustrates the possibility of learning rich embodied information of object, one step toward linking neural networks with symbolic representation in general. We expect future works to develop self-supervised learning models for natural categories beyond simple object identity, building on our work.

**Limitation:** in our experiment, object spatial location is inferred more easily than object pose (which we have not fully investigated), thus the predicted warping relies more on object translation than rotation. As almost all existing object-centric representation works, we focus on rigid bodies and simple environment. Future works need to explore how to learn object representation for deformable objects, objects with more complex shapes and lighting conditions, and more cluttered environment, towards more realistic application. **Potential negative societal impact**: as AI agents acquire more abilities that human beings have, including common knowledge of objects, there can be potential risk of misusing the technology for inhumane practice. The current capability of the model is still far from that stage but caution needs to be taken to regulate the usage of future development based on our work.

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

# A  Appendix

## A.1  Pseudo code of the OPPLE framework

---
**Algorithm 1** Developing object-centric representation by predicting future scene

---
**Initialize:** Network parameters $\theta$

**Require:** images $\boldsymbol{I}^{(t-1)}, \boldsymbol{I}^{(t)} \in \mathbb{R}^{w \times h \times 3}$, self-motion $\boldsymbol{v}_{\text{obs}}^{(t-1)}, \omega_{\text{obs}}^{(t-1)}, \boldsymbol{v}_{\text{obs}}^{(t)}, \omega_{\text{obs}}^{(t)}$

**Ensure:** prediction $\boldsymbol{I}'^{(t+1)}$, segmentation $\boldsymbol{\pi}_{1:K+1}^{(t-1)}$, $\boldsymbol{\pi}_{1:K+1}^{(t)}$, objects' codes $\boldsymbol{z}_{1:K}^{(t-1)}$, $\boldsymbol{z}_{1:K}^{(t)}$, objects'
 locations and poses $\hat{\boldsymbol{x}}_{1:K}^{(t-1)}, \boldsymbol{p}_{\phi_{1:K}}^{(t-1)}, \hat{\boldsymbol{x}}_{1:K}^{(t)}, \boldsymbol{p}_{\phi_{1:K}}^{(t)}$
 **for** $\tau = \{t-1, t\}$ **do**
  scene code $e^{(\tau)} \leftarrow \text{U-NetEncoder}_{f_\theta}(\boldsymbol{I}^{(\tau)})$
  object code $\boldsymbol{z}_{1:K}^{(\tau)}$, location $\hat{\boldsymbol{x}}_{1:K}^{(\tau)}$, pose $\boldsymbol{p}_{\phi_{1:K}}^{(\tau)} \leftarrow \text{LSTM}_{f_\theta}(e^{(\tau)})$
  background code $\boldsymbol{z}_{K+1} = 0$
  depth $\boldsymbol{D}^{(\tau)} \leftarrow h_\theta(\boldsymbol{I}^{(\tau)})$
  segmentation mask $\boldsymbol{\pi}_{1:K+1}^{(\tau)} \leftarrow \text{Softmax}\,(\text{U-NetDecoder}_{f_\theta}(\boldsymbol{I}^{(\tau)}, \boldsymbol{z}_{1:K}^{(\tau)}), 0)$
 **end for**
 object matching scores $r_{kl} \leftarrow \text{RBF}(\boldsymbol{z}_k^{(t)}, \boldsymbol{z}_l^{(t-1)}), k, l \in 1:K+1$
 **for** $k \leftarrow 1$ to $K$ **do**
  object motion $\hat{\boldsymbol{v}}_{1:K}, \boldsymbol{\omega}_{1:K} \leftarrow r_{k,l}, \hat{\boldsymbol{x}}_k^{(t)}, \hat{\boldsymbol{x}}_l^{(t-1)}, \boldsymbol{p}_{\phi_k}^{(t)}, \boldsymbol{p}_{\phi_l}^{(t-1)}, l = 1:K+1$
  onject-specific optical flow$_k \leftarrow \hat{\boldsymbol{v}}_{1:K}, \boldsymbol{\omega}_{1:K}, \boldsymbol{v}_{\text{obs}}, \omega_{\text{obs}}, \boldsymbol{D}^{(t)}$
 **end for**
 $\boldsymbol{I}'^{(t+1)}_{\text{warp}} \leftarrow \text{Warp}(\boldsymbol{I}^{(t)}, \text{optical flow}_{1:K+1})$
 $\boldsymbol{I}'^{(t+1)}_{\text{imagine}} \leftarrow g_\theta(\boldsymbol{I}^{(t)} \odot \boldsymbol{\pi}_{1:K+1}^{(t)}, log(\boldsymbol{D}^{(t)}) \odot \boldsymbol{\pi}_{1:K+1}^{(t)}, \boldsymbol{v}_{\text{obs}}, \omega_{\text{obs}}, \hat{\boldsymbol{v}}_{1:K+1}, \hat{\boldsymbol{x}}_{1:K})$
 final image prediction: $\boldsymbol{I}'^{(t+1)} \leftarrow \boldsymbol{I}'^{(t+1)}_{\text{warp}}, \boldsymbol{I}'^{(t+1)}_{\text{imagine}}, \text{warping weights}$
 update parameters: $\theta \leftarrow \theta - \gamma \nabla_\theta [|\boldsymbol{I}'^{(t+1)} - \boldsymbol{I}^{(t+1)}|^2 + \text{regularization loss}]$

---

## A.2  Problem formulation

We denote a scene as a set of distinct objects and a background $\mathbb{S} = \{O_1, O_2, \ldots, O_K, B\}$, where $K$ is the number of objects in scene. At any moment $t$, we denote two state variables, the location and pose of each object relative to the perspective of an observer (camera), as $\boldsymbol{x}_{1:K}^{(t)}$ and $\boldsymbol{\phi}_{1:K}^{(t)}$, where $\boldsymbol{x}_k^{(t)}$ is the 3-d coordinate of the $k$-th object and $\phi_k^{(t)}$ is its yaw angle from a canonical pose, as viewed from the reference frame of the camera (for simplicity, we do not consider pitch and roll and leave the generalization to 3D pose to future works). At time $t$, given the location of the camera $\boldsymbol{o}^{(t)} \in \mathbb{R}^3$ and its facing direction $\alpha^{(t)}$, $\mathbb{S}$ renders a 2D image on the camera as $\boldsymbol{I}^{(t)} \in \mathbb{R}^{w \times h \times 3}$, where $w \times h$ is the size of the image. Our goal is to train a neural network that infers properties of objects given only a single image $\boldsymbol{I}^{(t)}$ as the sole input without external supervision:

$$\{\boldsymbol{z}_{1:K}^{(t)}, \boldsymbol{\pi}_{1:K+1}^{(t)}, \hat{\boldsymbol{x}}_{1:K}^{(t)}, \boldsymbol{p}_{\phi_{1:K}}^{(t)}\} = f_{\text{obj}}(\boldsymbol{I}^{(t)}) \tag{1}$$

Here, $\boldsymbol{z}_{1:K}^{(t)}$ is a set of view-invariant vectors representing the identity of each object $k$. "View-invariant" is loosely defined as $|\boldsymbol{z}_k^{(t)} - \boldsymbol{z}_k^{(t+\Delta t)}| < |\boldsymbol{z}_k^{(t)} - \boldsymbol{z}_l^{(t)}|$ for $k \neq l$ and $\Delta t > 0$ in most cases, i.e., the vector codes are more similar for the same object across views than they are different across objects. $\boldsymbol{\pi}_{1:K+1}^{(t)} \in \mathbb{R}^{(K+1) \times w \times h}$ are the probabilities that each pixel belongs to any of the objects or the background ($\sum_k \pi_{kij} = 1$ for any pixel at $i, j$), which achieve object segmentation. To localize objects, $\hat{\boldsymbol{x}}_{1:K}^{(t)}$ are the estimated locations of each object relative to the observer and $\boldsymbol{p}_{\phi_{1:K}}^{(t)}$ are the estimated probability distributions of the poses of each object. Each $\boldsymbol{p}_{\phi_k}^{(t)} \in \mathbb{R}^b$ is a probability distribution over $b$ equally-spaced bins of yaw angles in $(0, 2\pi)$.

## A.3  Network architecture

The flow of information in networks is displayed in Figure 1.

**Object Extraction Network.** We build the Object Extraction Network $f_{\theta_{\mathrm{obj}}}$ by modification of a U-Net (Ronneberger et al., 2015). A basic U-Net is composed of a convolutional encoder and a transposed convolutional decoder, while each encoder layer sends a skip connection to the corresponding decoder layer, so that both global and local information can be combined in the decoder. Inside our $f_{\theta_{\mathrm{obj}}}$, an image $\boldsymbol{I}^{(t+1)}$ first passes through the encoder. Additional Atrous spatial pyramid pooling layer (Chen et al., 2017) is inserted between the middle two convolutional layers of the encoder to expand the receptive field. The top layer of the encoder outputs a feature vector $e^t$ capturing the global information of the scene. An LSTM further repeatedly reads in $e^{(t)}$ and sequentially outputs one vector for an object at a time. Each vector is then mapped through a one-layer fully connected network to predict object code $\boldsymbol{z}_k^{(t)}$, object location $\hat{\boldsymbol{x}}_k^{(t)}$ and object pose probability $\boldsymbol{p}_{\phi_k^{(t)}}$, $k = 1, 2, \cdots, K$. The location prediction is represented by polar angle and distance from the object and both are restricted to the range of possible value within the virtual environment by passing a scalar prediction through a logistic function and scaled and shifted to the corresponding ranges. The probability $\boldsymbol{p}_{\phi_k^{(t)}}$ of the pose falling in each bin is represented as $\log \boldsymbol{p}_{\phi_k^{(t)}}$ for numerical stability. Each object code vector $\boldsymbol{z}_k^{(t)}$ is then independently fed through the decoder with shared skip connections from the encoder. The decoder outputs one channel for each pixel, representing an un-normalized log likelihood that the pixel belongs to the object $k$. The unnormalized logit maps for all objects are concatenated with a map of all zero for the background, and compete through a softmax function pixel-wise to output the probabilistic segmentation map $\boldsymbol{\pi}_k^{(t)}$.

**Depth perception network.** We use a standard U-Net for depth perception function $h_{\theta_{\mathrm{depth}}}$ that processes images $\boldsymbol{I}^{(t)}$ and output a single-channel depth map $\boldsymbol{D}^{(t)}$.

**Object-based imagination network.** We build the imagination network $g_{\theta_{\mathrm{imag}}}$ also with a modified U-Net. The input is concatenated image $\boldsymbol{I}^{(t)}$ and log of depth $log(\boldsymbol{D}^{(t)})$ inferred by the depth perception network, both multiplied pixel-wise by one probabilistic mask $\boldsymbol{\pi}_k^{(t)}$ corresponding to each inferred object and the background. The output of the encoder network is concatenated with a vector composed of the observer's moving velocity $\boldsymbol{v}_{\mathrm{obs}}^{(t)}$ and rotational speed $\omega_{\mathrm{obs}}^{(t)}$, and the estimated object location $\hat{\boldsymbol{x}}_k^{(t)}$, velocity $\hat{\boldsymbol{v}}_k^{(t)}$ and rotational speed $\hat{\omega}_k^{(t)}$ inferred by the Object Extraction Netowrk, before entering the decoder. The decoder outputs five channels for each pixel: three for predicting the RGB colors, one for depth and one for the probability of the pixel belonging to any object $k$ or background and is used to weight the predicted color and depth for the final "imagination".

### A.4 Learning object representation by predicting the future

### A.4.1 Prediction by warping

We first describe the prediction of part of the next image by warping the current image. Here we consider only rigid objects and the fates of all visible pixel belonging to an object. With depth $\boldsymbol{D}^{(t)} \in \mathbb{R}^{w \times h} = h_\theta(\boldsymbol{I}^{(t)})$ of all pixel in a view inferred by the Depth Perception network based on visual features in the image $\boldsymbol{I}^{(t)}$, the 3D location of a pixel at any coordinate $(i, j)$ in the image, where $|i| \leq \frac{w-1}{2}, |j| \leq \frac{h-1}{2}$, can be determined given the focal length $d$ of the camera (in the unit of pixel size) as $\hat{\boldsymbol{m}}_{(i,j)}^{(t)} = \frac{D_{(i,j)}^{(t)}}{\sqrt{i^2 + j^2 + d^2}} \cdot [i, d, j]$. Here, we take the coordinate of the center of an image as (0,0). On the other hand, with $\hat{\boldsymbol{x}}_k^{(t)}$ and $\hat{\boldsymbol{x}}_k^{(t-1)}$, the current and previous locations of the object $k$ that the pixel $(i, j)$ belongs to that are inferred from $\boldsymbol{I}^{(t)}$ and $\boldsymbol{I}^{(t-1)}$ respectively, we can estimate the instantaneous velocity of the object $\hat{\boldsymbol{v}}_k^{(t)} = \hat{\boldsymbol{x}}_k^{(t)} - \hat{\boldsymbol{x}}_k^{(t-1)}$ (after correcting for camera motion). Similarly, with the inferred the current and previous pose probabilities of the object, $\boldsymbol{p}_{\phi_k}^{(t)}$ and $\boldsymbol{p}_{\phi_k}^{(t-1)}$, we can obtain the likelihood of its angular velocity

$$p(\phi_k^{(t)}, \phi_k^{(t-1)} \mid \omega_k^{(t)} = \omega) \propto \sum_{\substack{\gamma_1, \gamma_2, \gamma_1 - \gamma_2 \in \\ \{\omega - 2\pi, \omega, \omega + 2\pi\}}} p(\phi_k^{(t)} = \gamma_1) \cdot p(\phi_k^{(t-1)} = \gamma_2).$$ By additionally

imposing a prior distribution (we use Von Mises distribution) over $\omega_k^{(t)}$ that favors slow rotation, we can obtain the posterior distribution of the object's angular velocity $p(\omega_k^{(t)} \mid \phi_k^{(t)}, \phi_k^{(t-1)})$, and eventually the posterior distribution of the object's next pose $p(\phi_k^{(t+1)} \mid \phi_k^{(t)}, \phi_k^{(t-1)})$.

Assuming a pixel $(i,j)$ belongs to object $k$, using the estimated motion information $\hat{\boldsymbol{v}}_k^{(t)}$ and $p(\omega_k^{(t)} \mid \phi_k^{(t)}, \phi_k^{(t-1)})$ of the object, together with the current location and pose of the object and the current 3D location $\hat{\boldsymbol{m}}_{(i,j)}^{(t)}$ of the pixel, we can predict the 3D location $\boldsymbol{m'}_{k,(i,j)}^{(t+1)}$ of the pixel at the next moment as $\boldsymbol{m'}_{k,(i,j)}^{(t+1)} = \boldsymbol{M}_{-\omega_{\text{obs}}}^{(t)}[\boldsymbol{M}_{\hat{\omega}_k}^{(t)}(\hat{\boldsymbol{m}}_{(i,j)}^{(t)} - \hat{\boldsymbol{x}}_k^{(t)}) + \hat{\boldsymbol{x}}_k^{(t)} + \hat{\boldsymbol{v}}_k^{(t)} - \boldsymbol{v}_{\text{obs}}^{(t)}]$, where $\boldsymbol{M}_{-\omega_{\text{obs}}}^{(t)}$ and $\boldsymbol{M}_{\hat{\omega}_k}^{(t)}$ are rotational matrices due to the rotation of the observer and the object, respectively, and $\boldsymbol{v}_{\text{obs}}^{(t)}$ is the velocity of the observer (relative to its own reference frame at $t$). In this way, assuming objects move smoothly most of the time, if the self motion information is known, the 3D location of each visible pixel can be predicted. If a pixel belongs to the background, $\omega_{K+1} = 0$ and $\boldsymbol{v}_{K+1} = 0$ ($K+1$ is the background's index). Given the predicted 3D location, the target coordinate $(i', j')_k^{(t+1)}$ of the pixel on the image and its new depth $D'_k(i,j)^{(t+1)}$ can be calculated. This prediction of pixel movement allows predicting the image $\boldsymbol{I'}^{(t+1)}$ and depth $\boldsymbol{D'}^{(t+1)}$ by weighting the colors and depth of pixels predicted to land near each pixel on the discrete grid of the next frame, as explained below.

Because the object attribution of each pixel is not known but is inferred by $f_{\text{obj}}(\boldsymbol{I}^{(t)})$, it is represented for every pixel as a probability of belonging to each object and the background $\boldsymbol{\pi}_k^{(t)}$, $k = 1, 2, \cdots, K+1$. Therefore, the predicted motion of each pixel should be described as a probability distribution over $K+1$ discrete target locations $p(\boldsymbol{x'}_{(i,j)}^{(t+1)}) = \sum_{k=1}^{K+1} \pi_{kij}^{(t)} \cdot \delta(\boldsymbol{x'}_{k,(i,j)}^{(t+1)})$, i.e., pixel $(i,j)$ has a probability of $\pi_{kij}^{(t)}$ to move to location $\boldsymbol{x'}_{k,(i,j)}^{(t+1)}$ at the next time point, for $k = 1, 2, \cdots, K+1$. With such probabilistic prediction of pixel movement for all visible pixel $(i,j)^{(t)}$, we can partially predict the colors of the next image at the pixel grids where some original pixels from the current view will land nearby by weighting their contribution:

$$\boldsymbol{I'}_{\text{Warp}}^{(t+1)}(p,q) = \begin{cases} \frac{\sum_{k,i,j} w_k(i,j,p,q) I^{(t)}(i,j)}{\sum_{k,i,j} w_k(i,j,p,q)}, & \text{if } \sum_{k,i,j} w_k(i,j,p,q) > 0 \\ 0, & \text{otherwise} \end{cases} \tag{2}$$

We define the weight of the contribution from any source pixel $(i,j)$ to a target pixel $(p,q)$ as

$$w_k(i,j,p,q) = \pi_{kij}^{(t)} \cdot e^{-\beta \cdot D'_k^{(t+1)}(i,j)} \cdot \max\{1 - |i'_k^{(t+1)} - p|, 0\} \cdot \max\{1 - |j'_k^{(t+1)} - q|, 0\} \tag{3}$$

The first term incorporates the uncertainty of which object a pixel belongs to. The second term $e^{-\beta \cdot D'_k^{(t+1)}(i,j)}$ resolves the issue of occlusion when multiple pixels are predicted to move close to the same pixel grid by down-weighting the pixels predicted to land farther from the camera. This last two terms mean that only the source pixels predicted to land within a square of of $2 \times 2$ pixels centered at any target location $(p,q)$ will contribute to the color $I'_{\text{Warp}}^{(t+1)}(p,q)$. The depth map $\boldsymbol{D'}_{\text{Warp}}^{(t+1)}$ can be predicted by the same weighting scheme after replacing $I^{(t)}(i,j)$ with each predicted depth $D'_k^{(t+1)}(i,j)$ assuming the pixel belongs to object $k$.

### A.4.2 Imagination

For the regions not fully predictable by warping current image with equation (2), i.e., for $(p,q)$ where $\sum_{k,i,j} w_k(i,j,p,q) < 1$, we learn a function $g$ that "imagines" the appearance $\boldsymbol{I'}_{k\text{Imag}}^{(t+1)} \in \mathbb{R}^{w \times h \times 3}$ and the pixel-wise depth $\boldsymbol{D'}_{k\text{Imag}}^{(t+1)} \in \mathbb{R}^{w \times h}$ of the object or background $k$ in the next frame, and the predicted probabilities that each pixel in the next frame belongs to each object or the background $\boldsymbol{\pi'}_{k\text{Imag}}^{(t+1)} \in \mathbb{R}^{w \times h}$. The function takes as input portion of the current image corresponding to each object $\boldsymbol{I}^{(t)} \odot \boldsymbol{\pi}_k^{(t)}$ and the inferred depth $\boldsymbol{D}^{(t)} \odot \boldsymbol{\pi}_k^{(t)}$, both extracted by element-wise multiplying with the probabilistic segmentation mask $\boldsymbol{\pi}_k^{(t)}$, the information of the camera's self motion, and the location and motion of that object:

$$\{\boldsymbol{I'}_{k\text{Imag}}^{(t+1)}, \boldsymbol{D'}_{k\text{Imag}}^{(t+1)}, \boldsymbol{\pi'}_{k\text{Imag}}^{(t+1)}\} = g(\boldsymbol{I}_i^{(t)} \odot \boldsymbol{\pi}_k^{(t)}, \boldsymbol{D}^{(t)} \odot \boldsymbol{\pi}_k^{(t)}, \hat{\boldsymbol{x}}_k^{(t)}, \hat{\boldsymbol{v}}_k^{(t)}, \hat{\omega}_k^{(t)}, \omega_{\text{obs}}^{(t)}) \tag{4}$$

The "imagination" specific for each object and the background can then be merged using the weights prescribed by $\boldsymbol{\pi'}_{1:K\text{Imag}}^{(t+1)}$: $\boldsymbol{I'}_{\text{Imag}}^{(t+1)} = \sum_k \boldsymbol{I'}_{k\text{Imag}}^{(t+1)} \odot \boldsymbol{\pi'}_{k\text{Imag}}^{(t+1)}$, and $\boldsymbol{D'}_{\text{Imag}}^{(t+1)} = \sum_k \boldsymbol{D'}_{k\text{Imag}}^{(t+1)} \odot \boldsymbol{\pi'}_{k\text{Imag}}^{(t+1)}$.

### A.4.3 Combining warping and imagination

The final predicted image or depth map are weighted average of the prediction made by warping the current image or predicted depth map and the corresponding predictions by imagination:

$$\boldsymbol{I}'^{(t+1)} = \boldsymbol{I}'^{(t+1)}_{\text{Warp}} \odot \boldsymbol{W}_{\text{Warp}} + \boldsymbol{I}'^{(t+1)}_{\text{Imag}} \odot (1 - \boldsymbol{W}_{\text{Warp}}) \tag{5}$$

Here, $\boldsymbol{W}_{\text{Warp}} \in \mathbb{R}^{w \times h}$ with each element $W_{\text{Warp}}(p, q) = \max\{\sum_{k,i,j} w(i, j, p, q), 1\}$. The intuition is that imagination is only needed when there is not sufficient contribution for predicting a pixel by warping. The same weighting applies for generating the final predicted depth $\boldsymbol{D}'^{(t+1)}$.

In addition to the image, the states of the objects can also be predicted. The location of object $k$ at $t + 1$ can be predicted as $\boldsymbol{x}'^{(t+1)}_k = \boldsymbol{M}^{(t)}_{-\omega_{\text{obs}}}(\hat{\boldsymbol{x}}^{(t)}_k + \hat{\boldsymbol{v}}^{(t)}_k - \boldsymbol{v}^{(t)}_{\text{obs}})$. Its new pose probability can be predicted by $p'^{(t+1)}(\phi_k + \omega_{\text{obs}} = \gamma_2) = \sum\limits_{\substack{\gamma_1, \omega, \gamma_2 - \gamma_1 \in \\ \{\omega - 2\pi, \omega, \omega + 2\pi\}}} p(\hat{\omega}^{(t)}_k = \omega)p(\phi^{(t)}_k = \gamma_1)$ for $\gamma_2$ equal to each fixed yaw angle bin. To obtain $p'^{(t+1)}(\phi_k)$ instead of $p'^{(t+1)}(\phi_k + \omega_{\text{obs}})$ at the same set of bins, the vector $p'^{(t+1)}(\phi_k + \omega_{\text{obs}})$ can be shifted by multiplication with a pre-computed matrix composed of a bank of shifted Von Mises distributions to "move" the probability mess on the angle bins.

There is one important issue of object-centric representation when making prediction for future images: in order to predict the spatial state of each object at $t + 1$ based on the views at $t$ and $t - 1$, the network needs to match the representation of an object at $t$ from the representation of the same object at $t - 1$. As the dimensions of features (e.g., shape, surface texture, size, etc.) grows, the number of possible objects grows exponentially. Therefore, we cannot simply match object representations based on the order by which an LSTM extracts objects, as this requires learning a consistent order over enormous amount of objects. Instead, we take a soft-matching approach: we take a subset of the features in $\boldsymbol{z}^{(t)}_k$ extracted by $f$ as an identity code for each object. For object $k$ at time $t$, we calculate the distance between its identity code and those of all objects and the background at $t - 1$, and pass the distances through a radial basis function to serve as a matching score $r_{kl}$ indicating how closely the object $k$ matches each of the previous objects. The scores are used to weight all the estimated translational and rotational speeds for object $k$ each assuming a different object $l$ were the true object $k$ at $t - 1$. We additionally set the identity code of the background to be $\boldsymbol{z}_{K+1} = 0$, and its predicted motion to be zero.

### A.4.4 Learning objective

Above, we have explained how the next image input $\boldsymbol{I}'^{(t+1)}$, the depth map $\boldsymbol{D}'^{(t+1)}$ and the spatial states of each object, $\boldsymbol{x}'^{(t+1)}_k$ and $p'^{(t+1)}_{\phi_k}$ can be predicted based on object-centric representation extracted by a function $f$ from the current and previous images $\boldsymbol{I}^{(t)}_i$ and $\boldsymbol{I}^{(t-1)}_i$, the depth $\boldsymbol{D}'^{(t)}$ extracted by a function $h$, combined with the prediction from object-based imagination function $g$ that are all to be learned. Among the three prediction targets, only the ground truth of visual input $\boldsymbol{I}^{(t+1)}$ is available, while the other can only be inferred by $f$ and $h$ from $\boldsymbol{I}^{(t+1)}$. Therefore, for the prediction targets other than $\boldsymbol{I}^{(t+1)}$, we use the self-consistent loss between the predicted value based on $t$ and $t - 1$ and the inferred value based on $t + 1$ as additional regularization terms to constrain the functions $f$ and $g$.

To learn the functions $f$, $g$ and $h$, we approximate them with deep neural networks with parameters $\theta$ and optimize $\theta$ to minimize the following loss function:

$$L = L_{\text{image}} + \lambda_{\text{depth}} L_{\text{depth}} + \lambda_{\text{spatial}} L_{\text{spatial}} + \lambda_{\text{map}} L_{\text{map}} \tag{6}$$

Here, $L_{\text{image}} = \text{MSE}(I'^{(t+1)}, I^{(t+1)})$ is the image prediction error. $L_{\text{depth}} = \text{MSE}(\log(D'^{(t+1)}), \log(\hat{D}^{(t+1)}))$ is the error between the predicted and inferred depth. $L_{\text{spatial}} = \sum_{k=1}^{K} |\sum_{l=1}^{K+1} r_{kl} \boldsymbol{x}'^{(t+1)}_l - \hat{\boldsymbol{x}}^{(t+1)}_k|^2 - \sum_{k=1}^{K} \min\{|\hat{\boldsymbol{x}}^{(t+1)}_{\text{rand}} - \hat{\boldsymbol{x}}^{(t+1)}_k|, \delta\} + \sum_{k=1}^{K} |\hat{\boldsymbol{x}}^{(t+1)}_k - \sum_{i,j} \boldsymbol{m}^{(t+1)}_{i,j} \pi_{kij}|^2 + \sum_{k=1}^{K} D_{\text{KL}}(\hat{\boldsymbol{p}}^{(t+1)}_{\phi_k} || \sum_{l=1}^{K+1} r_{kl} \boldsymbol{p}'^{(t+1)}_{\phi_l})$ is the self-consistent loss on spatial information prediction. The first term is the error between inferred and predicted location of each object, while the calculation of the predicted location incorporates soft matching between objects in

consecutive frames. The second term is the negative term of contrastive loss, which prevents the network from reaching a local minimum where all objects are inferred at the same location relative to the camera. The third term penalizes the discrepancy between the inferred object location and the average location of pixels in its segmentation mask. The last term is the KL-divergence between the predicted and inferred pose for each object at $t + 1$. $L_{\text{map}} = \text{ReLu}(10^{-4} - \boldsymbol{\pi}_{1:K}) + \boldsymbol{\pi}_k \cdot \boldsymbol{\pi}_l$, for $k \neq l$ avoids loss of gradient when some pixels are assigned zero probability to some objects and discourages overlap between maps of different objects.

### A.5 Training details

For training we used mini-batches of 24 triplets and trained the networks over the dataset for 40 epochs. For testing we used 4000 images. We trained our algorithm on one Nvidia RTX6000 with 24GB VRAM. We selected learning rate of 3e-4 and regularization parameters $\lambda_{\text{spatial}} = 1.0$, $\lambda_{\text{depth}} = 0.1$ and $L_{\text{map}} = 0.005$ after incremental iteration over a small set of values in log scale.

### A.6 Dataset details

We procedurally generated dataset composed of 306445 triplets of images captured by a virtual camera with field of view of 90 degrees in a square room, using the software Unity. The camera translates horizontally and pans with random small steps between consecutive frames to facilitate the learning of depth perception. 3 objects with random shape, size, surface color or textures are spawned at random locations in the room and each move with a randomly selected constant velocity and panning speed. The translation and panning of the the camera is known to the networks. No other ground truth information is provided. The first two frames serve as data and the last frame serve as the prediction target at $t + 1$. An important difference between this dataset and other commonly used synthetic datasets for OCRL is that more complex and diverse textures are used on both the objects and the background.

### A.7 Object segmentaiton comparison setup

To compare our work with the states-of-art models of unsupervised object-centric representation learning, we trained MONet (Burgess et al., 2019), slot-attention (Locatello et al., 2020) and GENESIS[1] (Engelcke et al., 2021) on the same dataset. To address the concern that the original configurations of the models are not optimized for more difficult dataset, we trained variants of some of the models with large network size. For MONet, we tested channel numbers of [32, 64, 128, 128] (MONet-128) and [32, 64, 128, 256, 256] (MONet-128-bigger) for the hidden layers of encoder of the component VAE in addition to [32, 32, 64, 64] of the original paper and adjusted decoder layers sizes accordingly, and increased the base channel from 64 to 128 for the attention network. For slot attention, we tested both the original configuration and a variant which increased the number of features in the attention component from 64 to 128 (slot-attention-128). Slot numbers were chosen as 4 except for GENESIS. We adapted the following implementations of other models in our comparison:
1. MONET: https://github.com/stelzner/monet
2. Genesis: https://github.com/applied-ai-lab/genesis
3. Slot Attention: https://github.com/evelinehong/slot-attention-pytorch/

---

[1]We failed to to obtain reasonable result by training GENESIS V2 on our dataset, thus we adopted a GENESIS network pre-trained on GQN dataset and retrained on our dataset with K=7.

