# OpenReview forum: "Learning to perceive objects by prediction"
_NeurIPS.cc/2021/Workshop/SVRHM — SVRHM 2021 Oral_

### Official Review · Reviewer_iY7d · 2021-10-22
**Interesting self-supervised learning technique for objects in a scene**

**Rating:** 7
**Confidence:** 3

**Review:**

This work uses the prediction error in a series of frames to learn to extract objects without labels. The model consists of three subnetworks for depth estimation, object extraction and an "occlusion predictor" which are all trained jointly on the reconstruction error. The combination of these networks, which uses an explicit warping step to make use of some 3D knowledge of the scene from the depth estimator, is a novel contribution.

The manuscript is not super polished (e.g. some words are missing) but is understandable. The experiments (object segmentation and localization) are run on a synthetic datasets with some similarities to MONet. Competing methods like MONet perform surprisingly badly. I'd loved to see a setting in which MONet performs well, and to have a deeper discussion why competing methods fail on the proposed dataset.

---

### Official Review · Reviewer_Cyxs · 2021-10-31
**Great submission. Minor suggestions on how to place the work in context of past studies at the intersection of neuroscience and machine learning.**

**Rating:** 9
**Confidence:** 4

**Review:**

This is a strong submission, and I appreciate how the clarity which with the authors step through the paper from background, motivation, methods, and discussion. I also would like to thank the authors for carefully explain each component of the OPPLE network via a schematic figure and via text in the main paper as well as in the supplemental section. One conceptual/high-level comment that I kept popping up as I was reading through the manuscript concerns the connection between the proposed framework and the brain. The authors referred to the findings of object permanence in children and broadcasting of motor signals in sensory areas, yet, it is unclear how the architectures of the OPPLE network are setup to reflect these biological properties. Additionally, do there exist some empirical findings that confirm that some computations adopted in the proposed OPPLE network are also present in the brain? If so, are such computations specific to local cortical regions or are they global properties of the general brain-wide network?

---

### Official Review · Reviewer_3b4r · 2021-10-31
**A step towards a solution to a fundamental problem**

**Rating:** 8
**Confidence:** 4

**Review:**

The paper addresses a fundamental tension between human vision and most machine vision: even in young infants our vision seems "object-based"; we track objects rather than pixels, and have strong priors towards objects preserving their boundary shapes and movement trajectories. Somehow we must learn about the behaviours of these objects through mostly unsupervised means. Yet current unsupervised models (e.g. pixel-wise video prediction networks) fail to learn such objecthood-priors (e.g. they easily predict objects falling and drifting apart at moments of trajectory uncertainty).

The current paper doesn't yet propose an explanation for how such object priors might be learned or hardwired into our visual systems, but does present a working model that assumes the existence of objects, and from thereon learns about their locations, trajectories, and visual features through unsupervised predictive learning. The model consists of three streams of convolutional networks, all of which are trained concurrently to minimise image-wise prediction error on triplets of video sequences where both the camera and object can move, in rigid fashions. One stream outputs a depth map for each pixel in the scene, another outputs "object masks" indicating which pixels are occupied by which of a number of inferred objects, and the third of which makes a prediction of what occluded parts of the background or object might look like at the next moment. Networks also output scalar estimates of each object's pose and velocity. These outputs are then combined, by warping the image according to the object mask and object trajectory estimates (e.g. shifting the pixels marked as belonging to one object along its estimated trajectory), and combined with the "imagined" appearance of parts behind occlusions, to create a final prediction of the coming frame.

These several hand-engingeered steps are presumably not intended as a biologically plausible implementation, and they don't address the question of how a visual system learns to develop something like object masks in the first place. But they pose an effective solution to the problem of inferring object properties and making predictions on the basis of them. The system is currently constrained to discrete objects moving along rigid trajectories; it could not cope, for example, with the much fuzzier delineations in natural vision between objects, environment, and "stuff" like water, foliage, flocks of birds etc. But its self-supervised object segmentations and predictions are very impressive compared to the alternative models shown (which are perhaps not tailored to this test environment?), and it is necessary to begin with many constraints.

---

### Official Review · Reviewer_3Nmf · 2021-11-01
**Learning to perceive objects by prediction**

**Rating:** 9
**Confidence:** 4

**Review:**

# Quality & Clarity: 9

Abstract and introduction are compelling an motivating.
The authors clearly designate the objectives of the work centered around object-centric representational learning:
1. Develop a dataset consisting of moving objects with various visual properties that would challenge existing non-object centric models
2. Provide a novel implementation of object-centric models using 3D, tracked representations.

The model and problem formulations in 2.1 were clear. The authors use a collection three of modules to build an innovative architecture that can predict future frames given a learned 3D, tracked representations without labelled data. This model is well illustrated in Fig1.

Perhaps a personal note (and is relatively minor given the work as a whole), I found the third module's name "Imagination" to be distracting. In the work, it is described as predicting pixels attributed to objects that are about to be released from occlusion. While terms like "De-occlusion" are much less dramatic, they are more in line with the modules purpose in the current work. "Imagination" tends to bring about notions of counterfactual simulation etc.

The achievements of such an approach in terms of performance are well illustrated in Fig. 2 and Table 1.

In addition, the authors quantitatively evaluate that their architecture indeed learns to accurate localize objects and that these object representations contain a variance pattern that supports the hypothesis that the model has learned to de-confound shape, texture, and location.


# Originality 8

Currently in computer vision, as well-described by the authors, many approaches do no consider object representations as proper targets for learning. While not the first model using predictive learning, this work serves as one the first concrete examples from that field to focus on an object centric model (as opposed to a bigger, better, faster neural network) with discrete modules focusing on different aspects of 3D tracking.

# Significance 8

This work clearly shows the strength of un/self-supervised object-centric models under the framework of predictive learning. In this work, the authors showed that without such a perspective, other models suffer from confounding clearly distinguished information such as identity (ie shape, location, latent state) and location.

# Pros
- clearly describes problem and theoretical perspective
- illustrates a definite improvement in performance due to novel architecture
- illustrates that such improvements come from richer object-centric representations
- creates a novel dataset to emphasize key theoretical limitations to other non-predictive or non-object-centric approaches

# Cons
- The authors off limited insight as to how the fixed capacity of the architecture will generalize to scenes with variable objects or more complicated dynamics

---

### Decision · Program_Chairs · 2021-11-02

Accept (Oral)